# Effect of Microwave Treatment at 2.45 GHz on Soil Physicochemical Properties and Bacterial Community Characteristics in Phaeozems of Northeast China

Qi Li [1,2,†], Xiaohe Sun [1,2,3,†], Chunjiang Zhao [2,3,4], Shuo Yang [1,3], Chenchen Gu [1,3] and Changyuan Zhai [1,3,4,*]

1 Intelligent Equipment Research Center, Beijing Academy of Agriculture and Forestry Sciences, Beijing 100097, China
2 College of Resources and Environment, Jilin Agricultural University, Changchun 130118, China
3 Information Technology Research Center, Beijing Academy of Agriculture and Forestry Sciences, Beijing 100097, China
4 Nongxin (Nanjing) Smart Agriculture Research Institute Co., Ltd., Nanjing 211800, China
* Correspondence: zhaicy@nercita.org.cn; Tel.: +86-135-1917-3503
† These authors contributed equally to this work.

**Abstract:** Microwave irradiation is a new means of non-toxic, residue-free, and green soil disinfection that prevents and controls soil diseases, insects, and weeds and helps to improve crop quality and yield. Soil microorganisms, as an important part of the ecosystem, are closely related to crop growth and health. To investigate the changes of soil physicochemical properties and microbial communities during microwave soil disinfection for different time periods, phaeozems from northeastern China were selected for microwave treatment at 3, 6, 9, and 12 min, and their physicochemical properties were measured after 30 days of incubation. The test soils (0–20 cm) after 30 days of incubation were used, and high-throughput sequencing was performed to detect changes in their soil microbial structure under different microwave time treatments. Microwave treatment had significant effects on soil pH, nitrate ($NO_3^-$-N), ammonium ($NH_4^+$-N), and available phosphorus (AP) content. As shown by the Shannon, Chao, and Ace indices, microwave treatment at 3 min had the lowest effect on bacterial diversity compared to the control treatment (CK). Shannon index decreased by 9.92%, 24.56%, 34.37%, and 38.43% after 3, 6, 9, and 12 min microwave treatments, respectively; Chao index decreased by 7.69%, 18.13%, 32.21%, and 57.91%, respectively; Ace index decreased by 6.40%, 6.98%, 20.89%, and 52.07%, respectively. The relative abundance of beneficial soil microorganisms *Micromonospora*, *Fictibacillus*, *Paenibacillus,* and *Bacillus* (Firmicutes) increased significantly compared to CK. The results indicated that although the microwave treatment altered the soil microbial community, beneficial soil microorganisms showed faster recovery. In addition, pH, soil organic carbon (SOC), total nitrogen ratio (C/N), soil-available phosphorus (AP), and $NO_3^-$-N were important factors affecting bacterial community diversity and composition following microwave treatment, and bacterial community composition was driven by soil chemical properties such as soil pH, SOC, C/N, and $NO_3^-$-N. Microwave treatments at different time periods affected soil microbial community structure to different degrees, and soil bacteria of copiotrophic taxa (e.g., Firmicutes) were relatively higher than the control. Overall, microwave treatment from 3–6 min may be more suitable for soil disinfection. The study of the effect of microwave on soil physicochemical properties and bacterial microbial community not only provides some scientific reference for the rational application of microwave soil disinfection, but also has positive significance for soil-borne disease control and crop quality improvement.

**Keywords:** microwave; soil disinfection; soil physicochemical properties; soil bacteria



## 1. Introduction

Human mismanagement of agricultural production often results in adverse effects on the natural environment, such as soil diseases, insects, and weed infestations, all of which

greatly reduce the sustainability of farmland [1]. Soil disinfection is an important agronomic initiative to avoid soil infestation during agricultural production [2], yet traditional chemical soil disinfection may transfer chemical residues to neighboring ecosystems, leading to contamination and increased pest and weed resistance [3]. Microwave soil disinfection is expected to be a more environmentally friendly soil disinfection method with no pollution or drug residues.

Physical methods of soil disinfection such as the flame method, hot water method, and steam method mainly work through the thermal effect of denaturing proteins and inactivating pathogenic bacteria in the soil [4]. Unlike conventional heating for soil disinfection, microwave soil disinfection involves rapid oscillation of the polar molecules due to electromagnetic fields, which impacts polar macromolecular traits through such means as changing protease activity, affecting the charge distribution near the cell membrane of the organism, causing damage to the cell membrane producing membrane dysfunction, and interfering with or destroying normal cellular metabolic functions, leading to growth inhibition, cessation, or death of the organism [5]. The microwave soil disinfection process can lead to dielectric heating of water in the soil and organism's cells, and the value of the dielectric properties of soil varies depending on frequency, water content, density, and temperature [6–8]. When soil water content is the only variable, microwave action at deeper locations within the soil may become more difficult as soil water content and the loss factor increases; therefore, a suitable soil water content tends to be more favorable for the microwave process. Microwave action can produce dielectric heating of the soil and water in the cells of organisms; microwave irradiation of water-bearing soil under the best conditions for 1–4 min can quickly increase the soil temperature to 60–90 °C [9], achieving an optimal temperature range required to inhibit most grass seed activity. Most logarithmic soil pests, plant pathogenic fungi, and bacteria can reach inactivation conditions at 60 °C; when the soil temperature reaches 70 °C or higher, plant pathogenic fungi, bacteria, and most plant viruses are almost completely eliminated [10,11]. To be effective, the microwave process requires high energy intake by the soil [12]; nonetheless, microwaves have a similar "penetration" to chemical fumigation, which can directly affect the soil to a depth of several centimeters [13]. Soil physicochemical properties such as moisture content or texture are also important for microwave soil disinfection [14]; moreover, microwave irradiation can change the original soil physicochemical properties and soil microbial community in a manner that achieves soil disinfection. Microwave irradiation of soils is mainly reflected in thermal changes in the soil temperature [15]; however, the "non-thermal effect" of microwaves may also effectively influence soil biology [16]. The effect of microwave irradiation on soil microorganisms is highly similar to that of some chemical fungicides [17]; for example, both low-level microwave irradiation and chloroform fumigation of soil show an increase in extractable mineral nitrogen and a decrease in biomass [18], and this increase in nitrogen in both modes of action may originate from soil microorganisms [19].

Soil microorganisms are an important component of agroecosystems [20] and are vital for soil quality management and evaluation [21]. Microbial diversity is not only an important factor in maintaining soil function, but also a critical indicator for soil quality management and evaluation [22,23]. The soil biological community can basically restore the soil's original dynamic balance within 30 days following microwave irradiation, and soil microbial activity is closely related to soil physicochemical properties and has a specific function in nitrogen morphological transformation [24]. Therefore, increased soil nitrogen content after microwave action may be closely related to microbial community changes [25]. Soil bacterial diversity is the result of the combined effect of several soil environmental factors, among which, pH, organic matter, alkaline dissolved nitrogen, and available phosphorus and potassium content play an important role [26–29]. Soil bacteria can also participate directly or indirectly in soil biochemical processes, such as organic matter decomposition and synthesis, promotion of soil nutrient cycling and energy conversion [30], and the formation of organic matter, which improve soil structure to some extent [31]. Furthermore, soil bacteria play a key role in plant water and nutrient acqui-

sition, antagonizing soil-borne plant pests and pathogens, and inducing plant defense responses against pathogens [32]. Soil bacterial community composition is influenced by soil pH, organic matter content, and different agronomic practices [33,34]; thus, soil bacteria can be sensitive to changes brought about by the soil nutrient content and physicochemical properties [35]. In previous studies, the use of microwaves as a heat-driven soil disinfection method reduced the total bacterial community by nearly 90% after ~ 7 min of soil irradiation [36], and the duration of microwave irradiation determined the degree of soil bacterial inactivation.

Previous microwave soil irradiation studies have predominantly used microwave ovens with metal reflective cavity microwave transmitters for soil treatment [37]; when used with a metal cavity, the microwave process is bound to produce wave reflection, thus resulting in better utilization of the microwaves produced [38]. This changes the temperature distribution in the soil and thus affects the soil properties [39]. During microwave soil disinfection operations, microwaves are usually applied directly to the soil surface after passing through only the waveguide [40]. In order to fully understand the potential effect of microwave treatment on soil and its role in shaping the soil microbial community, a microwave generator without a metal reflector cavity was built, which is more suitable for microwave soil disinfection operations in the field. Phaeozems are clayey soils with strong expansion and contraction and disturbance characteristics. Phaeozems in northeast China are located in one of the world's major phaeozems belts [41], which are equivalent to the US classification of metamorphic soils and the UN classification of metamorphic soil units; this kind of soil has good natural conditions and a high fertility, making it very suitable for agricultural farming [42]. Therefore, the conservation and sustainable development of phaeozems have also become an issue worthy of attention. In this paper, we selected northeastern phaeozems with a high organic matter content for 3, 6, 9, and 12 min microwave irradiation tests. We hypothesized that microwave treatment would alter the soil bacterial community according to the temperature sensitivity of different taxa and that the damaged community would largely recover to its initial level within 30 days after the thermal disturbance. The physicochemical properties of soil naturally placed under greenhouse conditions for 30 days were analyzed to detect soil bacterial community composition and to investigate the effects of microwave soil disinfection on bacterial communities at different time periods, which may be important for the determination of soil system persistence and soil productivity.

## 2. Materials and Methods

### 2.1. Microwave Processing and Soil Sample Collection

The test soil was collected from the greenhouse of the teaching and research base of Jilin Agricultural University (125°24′44.136″ N, 43°49′23.2104″ E, Changchun, Jilin Province, China), which was loamy clay with 34.76% clay grains, 26.44% chalk grains, 37.07% sand grains, and a 1313.9 kg/m$^3$ density. The humus content of phaeozems in Northeast China tends to be higher than that in other types of soils [43]. After passing the soil samples through a 2-mm sieve, the wet basis moisture content of the collected soil was measured at 15.4% by the drying method, and the moisture content of the phaeozems was 10% when the optimum heat production conditions were obtained during microwave irradiation [44]. Thus, the soil moisture content was reduced by the natural air-drying method and the soil was loaded into clay pots (17 × 20 cm) for continuous microwave irradiation for 3, 6, 9, and 12 min. Microwave frequencies of 2.45 GHz and 915 MHz are generally suitable for industrial agriculture and households in China, and in terms of soil warming capacity, 2.45 GHz may be a more suitable choice if the soil is warmed to the same temperature [44]; for this reason, this study used microwave irradiation equipment with a microwave frequency of 2.45 GHz and a power of 2 kW; the main materials are shown in Table 1. The internal temperature of the soil after microwave treatment was measured by a PT100 sensor (Figure 1), which was inserted vertically into the soil to measure each centimeter of soil temperature in turn. The pots were numbered MW3, MW6, MW9, and

MW12 for the time periods 3, 6, 9, and 12 min, respectively, and the illuminated group without microwave treatment was labelled as CK. Each microwave time period and the control group had four replicate treatments. The soil area temperature after microwave irradiation for different time periods is shown in Table 2 [44], and the treated soil was placed in the National Experimentation for Precision Agriculture Research Demonstration Base No. 1 greenhouse (National Experimentation for Precision Agriculture, 40°10′31″ N, 116°26′10″E, Beijing, China) for 30 days of soil static management. The average temperature in the greenhouse during the 30 days was 27 °C and the humidity was 87.7%.

**Table 1.** Main equipment and materials of the microwave soil irradiation test bench.

| Equipment (Material) | Specification (Model) | Quantity | Manufacturer | Microwave Soil Irradiation |
|---|---|---|---|---|
| Magnetron | 2 M 362 | 1 | LG, Seoul, Korea | |
| Waveguide | 30 × 12.5 × 4.5 cm | 1 | MEGMEET Electric Co., Ltd., Shenzhen, China | |
| Microwave frequency conversion power supply | WepeX-1000B | 1 | MEGMEET Electric Co., Ltd., Shenzhen, China | |
| Microwave controller | WepeX-C1 | 1 | MEGMEET Electric Co., Ltd., Shenzhen, China | |
| Power display meter | D69-2049 | 1 | Elecall Electric Co., Ltd., Shenzhen, China | |
| Fixture | 100 × 60 × 60 cm | 1 | Self-made | |
| Movable lift platform | 46 × 32 × 20 cm | 1 | Karlon Hardware Store, Jiaxing, Zhejiang, China | |
| Temperature sensor | PT100 | 30 | Senxtee Electric Co., Ltd., Hangzhou, China | |
| Signal acquisition module | DS18B20 | 4 | Senxtee Electric Co., Ltd., Hangzhou, China | |

**Table 2.** Soil area temperature at different times of microwave irradiation.

| Treatment | Average Temperature at 20 cm Irradiation Depth/°C | Average Maximum Temperature of Layer/°C | Depth Where the Average Maximum Temperature of the Layer Is Located/cm | Modeling of Soil Temperature Inside Planters |
|---|---|---|---|---|
| CK | 0 | 0 | 0 | |
| MW3 | 44.9 | 59.4 | 4 | |
| MW6 | 58.1 | 80.2 | 5 | |
| MW9 | 66.1 | 89.1 | 4 | |
| MW12 | 74.5 | 100 | 4 | |

CK: control; MW3: microwave irradiation of the soil for 3 min; MW6: microwave irradiation of the soil for 6 min; MW9: microwave irradiation of the soil for 9 min; MW12: microwave irradiation of the soil for 12 min.

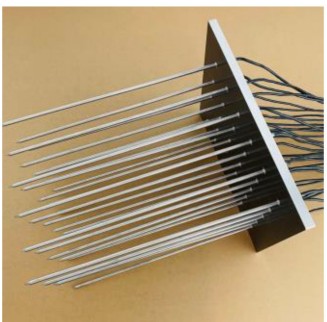

**Figure 1.** PT100 soil internal temperature acquisition device.

After 30 days of incubation, five random soil cores (0–20 cm deep and 2.2 cm in diameter) were collected from each pot and pooled together to form one composite sample per treatment. The sampling selection by multi-point mixing method via a straight-through-the-soil drill was used for sampling, for a total of ten soil samples (Figure 2). Plants, visible animals, and stone particles were removed from the samples and passed through a 2-mm sieve. Each composite soil sample was stored in three parts: the first part was snap frozen in liquid nitrogen and then transferred to a −80 °C refrigerator for DNA extraction and microbial community fractionation; the second part was air dried and stored at room temperature (25 °C) for the measurement of physical and chemical soil properties such as soil organic carbon (SOC), pH, etc.; the third part was stored in a −20 °C refrigerator for mineral nitrogen [ammonium ($NH_4^+$-N) and nitrate ($NO_3^-$-N)] measurement.

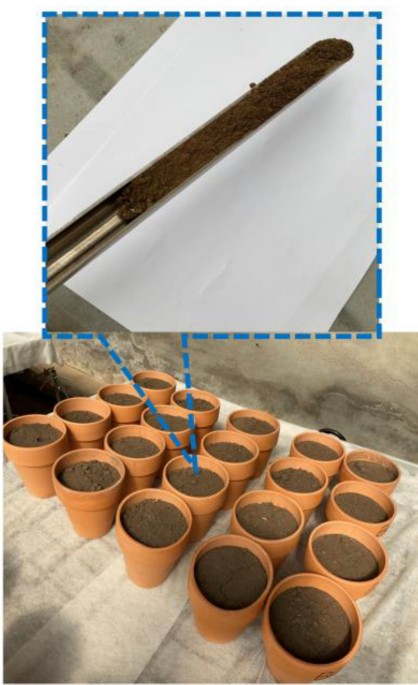

**Figure 2.** Soil sample incubation and sampling.

### 2.2. Soil Physicochemical Properties Analysis

Soil pH was determined using the potentiometric method; total nitrogen (TN) content was determined by the Vario EL III Elemental Analyzer (Elementar, Frankfurt, Hanau, Germany); SOC was determined by the potassium dichromate oxidation method. Soil-available phosphorus (AP) and available potassium (AK) were measured according to Hanway and Heidal's (1952) and Olsen's methods (1954) [45,46], respectively. $NH_4^+$-N and $NO_3^-$-N contents were analyzed by an AA3 continuous flow analyzer (AA3,SEAL Analytical, Shanghai, China) after extraction with 2 mol/L KCL.

### 2.3. Soil DNA Extraction and High-Throughput Sequencing

DNA was extracted from fresh soil (0.25 g) using the PowerSoil kit (MoBio Laboratories, Carlsbad, CA, USA) according to the manufacturer's instructions. The DNA extract was checked on 1% agarose gel and DNA concentration and purity were determined with a NanoDrop 2000 UV-vis spectrophotometer (Thermo Scientific, Wilmington, USA). The hypervariable region <u>V3-V4</u> of the bacterial 16S rRNA gene was amplified with primer pairs, 338F (5'-ACTCCTACGGGAGGCAGCAG-3') and 806R (5'-GGACTACHVGGGTWTCTAAT-3'), by an ABI GeneAmp® 9700 PCR thermocycler (ABI, Carlsbad, CA, USA). The PCR amplification of the 16S rRNA gene was performed as follows: initial denaturation at 95 °C for 3 min, followed by 27 cycles of denaturing at 95 °C for 30 s, annealing at 55 °C for 30 s and extension at 72 °C for 45 s, single extension at 72 °C for 10 min, and end at 4 °C. The PCR mixtures contain $5 \times 4$ μL *TransStart* FastPfu buffer, 2 μL 2.5 mM dNTPs, 0.8 μL 5 μM forward primer, 0.8 μL 5 μM reverse primer, 0.4 μL *TransStart* FastPfu DNA Polymerase, 10 ng template DNA, and finally sufficient ddH$_2$O to make up 20 μL. PCR reactions were performed in triplicate. The PCR product was extracted from 2% agarose gel and purified using the AxyPrep DNA Gel Extraction Kit (Axygen Biosciences, Union City, CA, USA) according to the manufacturer's instructions and quantified using a Quantus™ Fluorometer (Promega Corporation, Madison, Wisconsin, USA).

Quality control of the raw sequences was performed using fastp [47] (https://github.com/OpenGene/fastp, accessed on 6 December 2022, version 0.20.0) software and FLASH [48] (http://www.cbcb.umd.edu/software/flash, accessed on 6 December 2022, version 1.2.7) software for splicing: (i) the 300 bp reads were truncated at any site receiving an average quality score of <20 over a 50 bp sliding window and the truncated reads shorter than 50 bp as well as reads containing ambiguous characters were discarded; (ii) only overlapping sequences longer than 10 bp were assembled according to their overlapped sequence. The maximum mismatch ratio of the overlap region was 0.2. Reads that could not be assembled were discarded; (iii) samples were distinguished according to the barcode and primers and the sequence direction was adjusted for exact barcode matching with two nucleotide mismatches in primer matching. Operational taxonomic units (OTUs) with a 97% similarity cutoff [49,50] were clustered using UPARSE software (http://drive5.com/uparse/, accessed on 6 December 2022, version 7.1) [51] and chimeric sequences were identified and removed. The taxonomy of each OTU representative sequence was analyzed using an RDP Classifier (http://rdp.cme.msu.edu/, accessed on 6 December 2022, version 2.2) [52] against the 16S rRNA database (e.g., Silva v138) using a confidence threshold of 0.7.

### 2.4. Data Processing and Analysis

Statistical analysis was performed with SPSS 22.0 software (IBM Corp., Armonk, NY USA) using a one-way analysis of variance (ANOVA) procedure to test for normal distribution and homoscedasticity of soil physicochemical properties, bacterial diversity, and bacterial community composition. The least significant difference (LSD) test was used to compare the significance of differences between groups at the $p = 0.05$ level. Using the Vegan package, principal coordinate analysis (PCoA) based on the Bray–Curtis dissimilarity index showed the variation of bacterial communities in soil samples. The relationship between soil chemical properties and soil bacterial communities was analyzed using redundancy analysis (RDA). The relationship between soil chemical properties and bacterial community diversity and composition (at the phylum and genus level) was evaluated by Pearson correlation analysis, which was implemented with graphs through R software (version v.4.1.2) with the Hmisc and corrplot package (R Core Team, Auckland University, Auckland City, New Zealand 2021).

## 3. Results

### 3.1. Effect of Microwaves on Soil Physicochemical Properties at Different Time Periods

The changes observed in soil physicochemical properties after microwave treatment are presented in Table 3. Microwave treatment had significant effects on soil pH, $NO_3^-$-N, $NH_4^+$-N, and TN across the different time periods. $NO_3^-$-N increased by 13.7% and 7.6% after 3 and 12 min of microwave treatment, respectively, and decreased by 6.1% and 18.3% after 6 and 9 min of microwave treatment, respectively, compared with the control group, CK. $NH_4^+$-N increased by 115.3%, 58.0%, 12.6%, and 66.8% after 3, 6, 9, and 12 min microwave treatment, respectively; TN increased by 6.9%, 4.0%, 2.3%, and 4.6%, respectively, compared to CK; and AP increased by 19.7%, 13.4%, 12.0%, and 5.3%, respectively.

**Table 3.** Effect of microwave treatments on soil physicochemical properties at different time periods.

| Treatment | pH | SOC (g kg$^{-1}$) | TN (g kg$^{-1}$) | C/N | AP (mg kg$^{-1}$) | AK (mg kg$^{-1}$) | $NO_3^-$-N (mg kg$^{-1}$) | $NH_4^+$-N (mg kg$^{-1}$) |
|---|---|---|---|---|---|---|---|---|
| CK | 6.99 ± 0.00 c | 19.66 ± 0.11 a | 1.74 ± 0.01 b | 11.28 ± 0.09 a | 73 ± 0.56 c | 171±1.27 b | 109.33 ± 1.43 b | 23.80 ± 0.94 b |
| MW3 | 6.99 ± 0.01 c | 19.74 ± 0.08 a | 1.86 ± 0.01 a | 11.09 ± 0.06 ab | 88 ± 0.77 a | 183±0.20 a | 124.33 ± 0.41 a | 51.23 ± 0.50 a |
| MW6 | 7.07 ± 0.01 ab | 18.62 ± 0.63 ab | 1.81 ± 0.01 ab | 10.60 ± 0.33 ab | 83 ± 0.43 ab | 183 ± 1.54 a | 102.67 ± 0.41 ab | 37.60 ± 0.18 ab |
| MW9 | 7.12 ± 0.01 a | 19.70 ± 0.11 a | 1.78 ± 0.01 b | 10.30 ± 0.01 bc | 82 ± 0.72 abc | 170 ± 0.35 b | 89.37 ± 0.34 b | 26.80 ± 0.79 b |
| MW12 | 7.04 ± 0.00 bc | 17.69 ± 0.12 b | 1.82 ± 0.01 ab | 9.74 ± 0.12 c | 77 ± 0.49 bc | 173 ± 0.89 ab | 117.67 ± 0.82 a | 39.70 ± 0.64 ab |

Each value represents the mean ± standard error (n = 4). Letters within the same row indicate significant differences among different microwave treatments at $p < 0.05$. SOC: soil organic carbon; TN: total nitrogen; C/N: SOC/TN; AP: available phosphorus; AK: available potassium; $NO_3^-$-N: nitrate N; $NH_4^+$-N: ammonium N. CK: control; MW3: microwave irradiation of the soil for 3 min; MW6: microwave irradiation of the soil for 6 min; MW9: microwave irradiation of the soil for 9 min; MW12: microwave irradiation of the soil for 12 min.

### 3.2. Effect of Microwave Irradiation on Soil Bacterial Communities at Different Time Periods

#### 3.2.1. Effect of Microwaves on Soil OTU Numbers at Different Time Periods

OTU is a uniform marker artificially assigned to a taxonomic unit (strain, genus, species, group, etc.) in population genetics studies. To determine the number of overlapping OTUs and specific OTUs of soil bacteria under microwave treatments at different time periods, community analysis was performed using Veen plots (Figure 3); 1423 overlapping OTUs were present in the five treatments, accounting for 25.6% of the total number of OTUs. Among them, 4382 bacterial OTUs were unique to CK, 4146 were unique to MW3, 3598 were unique to MW6, 3190 were unique to MW9, and 2186 were unique to MW12.

The number of unique OTUs in CK treatment was 413, accounting for 7.4% of the total number of OTUs; 279 in MW3, accounting for 5.0%; 130 in MW6, accounting for 2.3%; 112 in MW9, accounting for 2.0%; and 118 in MW12, accounting for 2.1%. The number of OTUs specific to MW3, MW6, MW9, and MW12 was reduced by 32.4%, 68.5%, 72.8%, and 71.4%, respectively, compared with that of CK treatment; thus, microwave treatment reduced the number of unique OTUs.

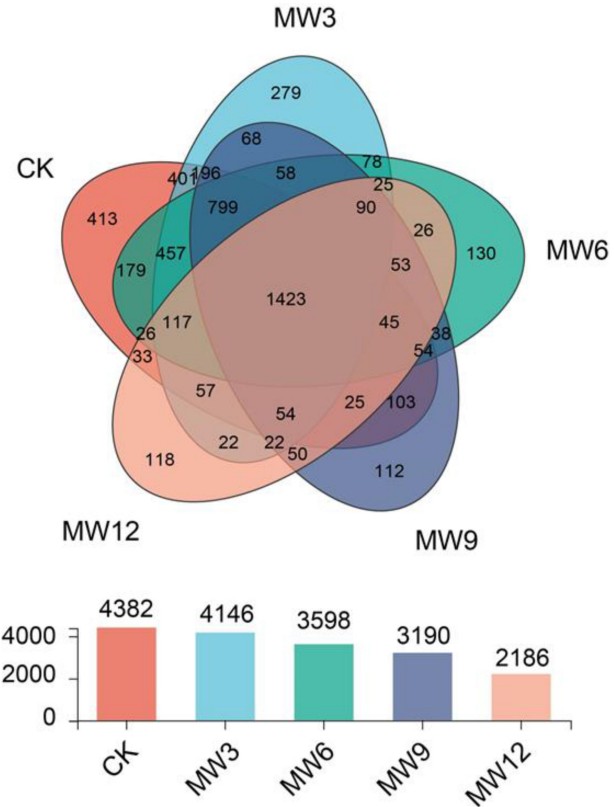

**Figure 3.** Venn Diagram showing unique and overlapped OTUs between the microwave treatments. CK: control; MW3: microwave irradiation of the soil for 3 min; MW6: microwave irradiation of the soil for 6 min; MW9: microwave irradiation of the soil for 9 min; MW12: microwave irradiation of the soil for 12 min.

3.2.2. Effect of Microwaves on Soil Bacterial Diversity at Different Time Periods

To comprehensively evaluate the alpha diversity of bacterial communities, the Shannon index was chosen to depict the diversity of these communities, and Chao and Ace indices were chosen to reflect the species richness. As shown in Figure 4, the microwave treatments at different time periods had significant effect on the Shannon index; among them, the Shannon index of the MW12 treatment was lower than that of the other treatments ($p < 0.05$; Figure 4a). Compared with CK treatment, the Shannon index of MW3, MW6, MW9, and MW12 treatments decreased by 9.92%, 24.56%, 34.37%, and 38.43%, respectively, indicating that the diversity of the soil bacterial community decreased with an increase in the microwave radiation time. The Chao indices decreased by 7.69%, 18.13%, 32.21%, and 57.91%, respectively. The Ace index decreased by 20.89% and 52.07% under MW9 and MW12 treatments, respectively ($p < 0.05$; Figure 4). Chao and Ace indices showed a trend of decreasing soil bacterial community diversity with increasing microwave time, indicating that microwave treatments affect bacterial community richness, and irradiation duration may be a key factor affecting bacterial community richness.

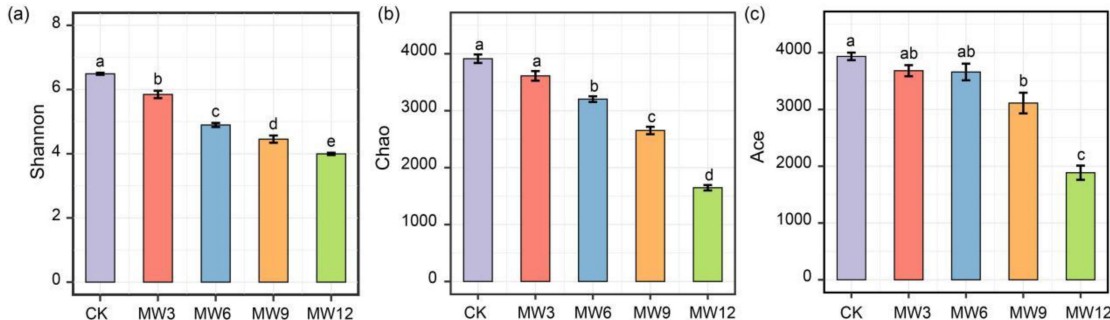

**Figure 4.** Effect of different microwave treatments on bacterial diversity shown via the (**a**) Shannon, (**b**) Chao, and (**c**) Ace indices. (mean ± SE, n = 4). Different lower-case letters indicate the significant differences of variable means among the different microwave time periods. CK: control; MW3: microwave irradiation of the soil for 3 min; MW6: microwave irradiation of the soil for 6 min; MW9: microwave irradiation of the soil for 9 min; MW12: microwave irradiation of the soil for 12 min.

### 3.3. Effect of Microwaves on Soil Bacterial Variation at Different Time Periods

The variation in the composition of the soil bacterial community following different microwave treatments was analyzed by PCoA at the OTU level (Figure 5). The results indicated that the bacterial communities of the MW3 and MW12 treatments differed from those in the CK treatment, while the communities from the MW6 and MW9 treatments were more similar in composition. In total, PCoA showed 85.79% variation in bacterial communities, with the PC1 axis accounting for 75.41% of the variation and the PC2 axis accounting for 10.38% of the variation. The four replicate samples for each treatment were clustered together indicating high reproducibility. The results indicated significant differences in bacterial community structure at the different microwave treatment times.

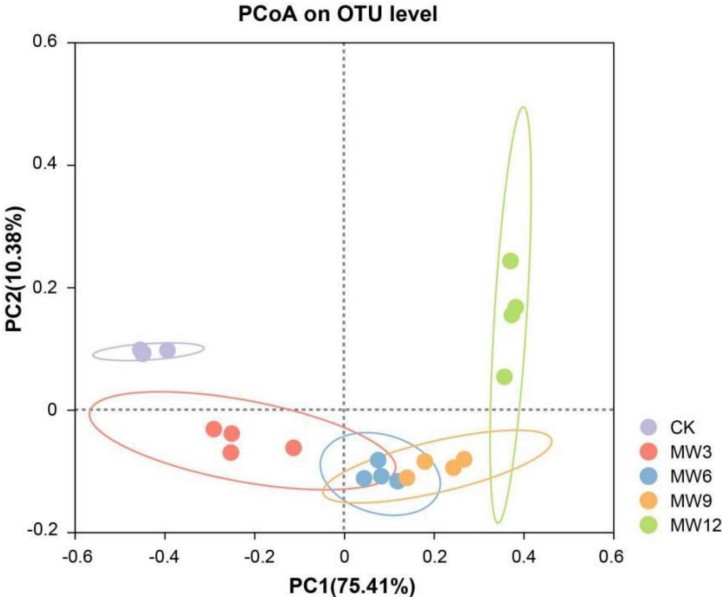

**Figure 5.** Principal coordinate analysis (PCoA) of the bacterial community structure based on Bray—Curtis distance. CK: control; MW3: microwave irradiation of the soil for 3 min; MW6: microwave irradiation of the soil for 6 min; MW9: microwave irradiation of the soil for 9 min; MW12: microwave irradiation of the soil for 12 min.

### 3.4. Effect of Microwaves on Soil Bacterial Community at the Phylum and Class Levels at Different Time Periods

The V3–V4 regions of bacterial 16S rRNA genes in soil samples after different microwave treatments were sequenced and classified (both unranked and unclassified). A

total of 37 phyla, 131 classes, 301 orders, 483 families, 942 genera, and 2022 species were detected in the 16 analyzed soil samples.

The taxonomic distribution of bacterial communities was evaluated at different classification levels and sequences in each OTU were analyzed with 97% similarity. As shown in Figure 6a, Actinobacteria (33%), Proteobacteria (17%), and Acidobacteria (15%) dominated in the CK treatment. Firmicutes and Actinobacteria were the dominant phyla in the microwave-treated soil, with the relative abundance percentages of Firmicutes at 3, 6, 9, and 12 min microwave time gradients being 28%, 58%, 69%, and 83%, respectively. Notably, the relative abundance of Proteobacteria and Actinobacteria decreased with the microwave time gradient. The relative abundance of Chiloroflexi and Acidobacteria was higher in the CK treatment. The relative abundance of Chiloroflexi decreased by 22%, 54.8%, 73.8%, and 92% after 3 min, 6 min, 9 min, and 12 min microwave treatments, respectively, and the relative abundance of Acidobacteria decreased by 62%, 69%, 77%, and 93%, respectively. The relative abundance of Acidobacteria decreased by 62%, 69%, 77%, and 93%, respectively.

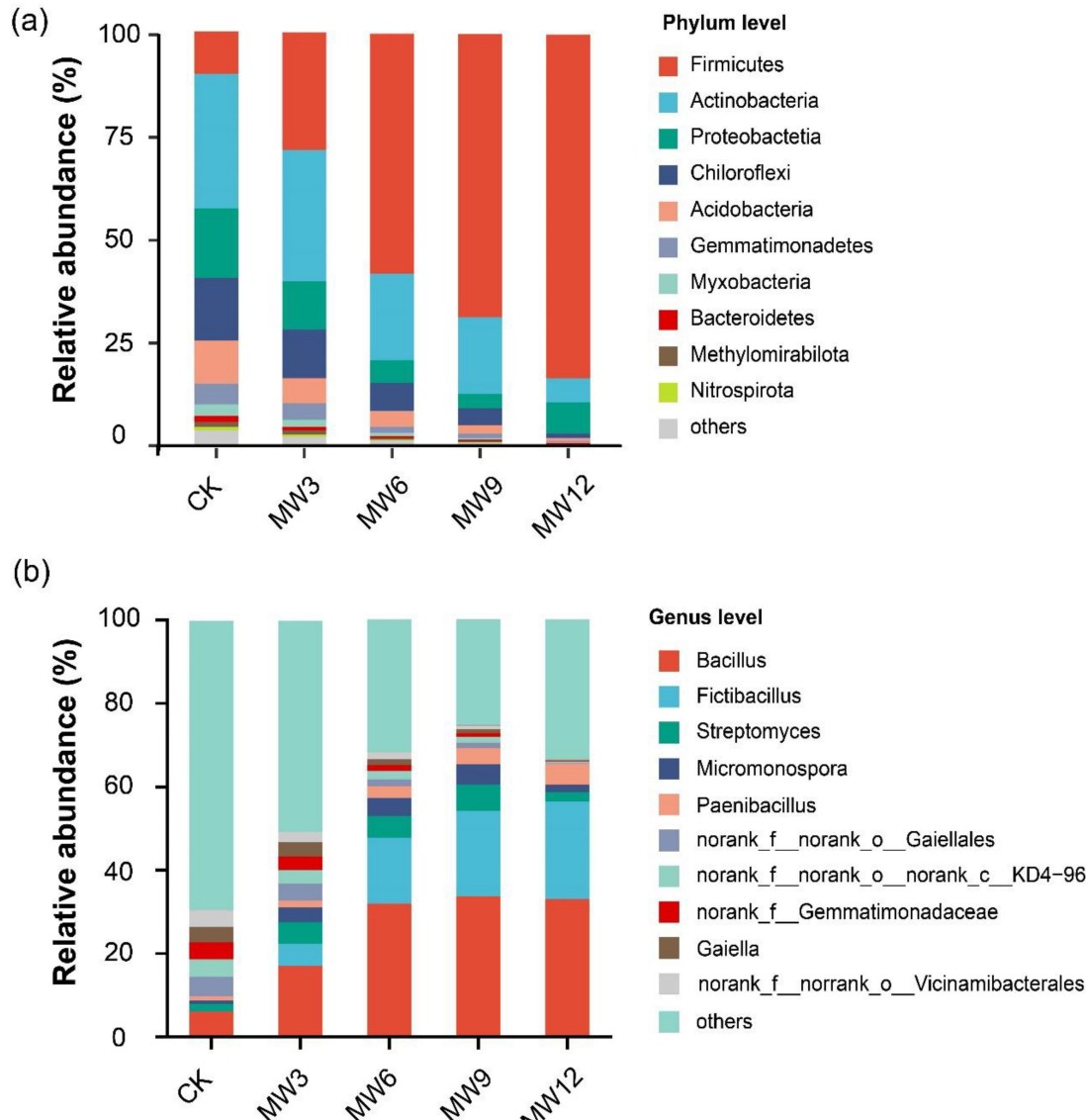

**Figure 6.** Relative abundance of soil bacteria in different microwave treatments at the phylum and genus level. Relative abundance at (**a**) phylum level; (**b**) genus level. CK: control; MW3: microwave irradiation of the soil for 3 min; MW6: microwave irradiation of the soil for 6 min; MW9: microwave irradiation of the soil for 9 min; MW12: microwave irradiation of the soil for 12 min.

At the genus level, as shown in Figure 6b, the top five genera with high relative abundance were *Bacillus*, *Fictibacillus*, *Streptomyces*, *Micromonospora*, and *Paenibacillus* belonging to Firmicutes and *Streptomyces* and *Micromonospora* belonging to Actinobacteria. Five genera accounted for 9.6%, 32.5%, 60%, 69.2%, and 65.4% of all genera in the CK, MW3, MW6, MW9, and MW12 treatments, respectively. Compared to the CK treatment, the relative abundance of *Bacillus* increased 1.8-, 4.4-, 4.6-, and 4.5-fold under MW3, MW6, MW9, and MW12 treatments, respectively. Similarly, the relative abundance of *Fictibacillus* increased with increasing microwave time, 118-, 355-, 460-, and 526-fold under MW3, MW6, MW9, and MW12 treatments, respectively, compared to that in the CK treatment. The relative abundance of *Streptomyces*, *Micromonospora*, and *Paenibacillus* was lower under the CK treatment and higher under microwave treatment at 3, 6, 9 and 12 min. Compared to CK treatment, the relative abundance of *Streptomyces* increased 1.9-, 1.85-, 2.5-, and 0.2-fold in the MW3, MW6, MW9, and MW12 treatments, respectively; the relative abundance of *Micromonospora* increased 3.4-, 4.5-, 5.1-, and 1.2-fold in MW3, MW6, MW9, and MW12 treatments, respectively. The relative abundance of *Paenibacillus* increased 0.6-, 1.7-, 2.8-, and 3.9-fold in MW3, MW6, MW9, and MW12 treatments, respectively.

### 3.5. Correlation of Soil Bacterial Communities with Environmental Factors

RDA analysis of community abundance at the level of bacterial OTUs was performed using CANOCO 5.0 software, and the maximum eigenvalue in the four axes was 2.64 > 3.5; thus, RDA in the linear model was selected. The effects of soil environmental factors on bacterial microbial communities are shown in Figure 7. Soil environmental factors were responsible for a total of 84.1% of the variation in soil bacterial communities, indicating that the RDA analysis successfully quantified the relationship between soil bacterial communities and soil chemical properties after microwave treatment, with the RDA1 and RDA2 axes accounting for 75.08% and 9.1% of the total variation, respectively (Figure 7). The MW6 and MW9 treatments were in the same quadrant and closer to the MW3 treatment, indicating some similarity in soil bacterial community abundance, while the MW12 treatment was farther away from the other treatments, indicating the uniqueness of its bacterial community structure. The length of the arrow pointing to the environmental factors expresses the magnitude of their influence on the bacterial community structure. The Mantel test further verified that the dominant factors affecting the soil bacterial community structure were soil pH ($r = 0.837$, $p = 0.001$), AP ($r = 0.743$, $p = 0.002$), SOC ($r = 0.640$, $p = 0.015$), and total nitrogen ratio (C/N; $r = 0.590$, $p = 0.024$), of which, C/N had a strong correlation with SOC (Table 4). Soil pH and AP were more strongly correlated with the MW6 and MW9 treatments, while soil C/N and SOC were more strongly correlated with the MW3 treatment (Figure 7).

**Table 4.** Spearman's rank correlation (R-value) between soil chemical properties and bacterial communities based on the Mantel test.

| Factors | r | p |
|---|---|---|
| pH | 0.838 | 0.001 |
| AP | 0.744 | 0.002 |
| SOC | 0.641 | 0.015 |
| C/N | 0.590 | 0.024 |
| $NO_3^-$-N | 0.506 | 0.082 |
| AK | 0.471 | 0.124 |
| TN | 0.301 | 0.444 |
| $NH_4^+$-N | 0.189 | 0.743 |

Significant correlation between bacterial community and soil chemical properties based on the Mantel test ($p < 0.05$). SOC: soil organic carbon; C/N: Soil organic carbon and total nitrogen ratio; TN: total nitrogen; AP: available phosphorus; AK: available potassium; $NH_4^+$-N: ammonium N; and $NO_3^-$-N: nitrate N.

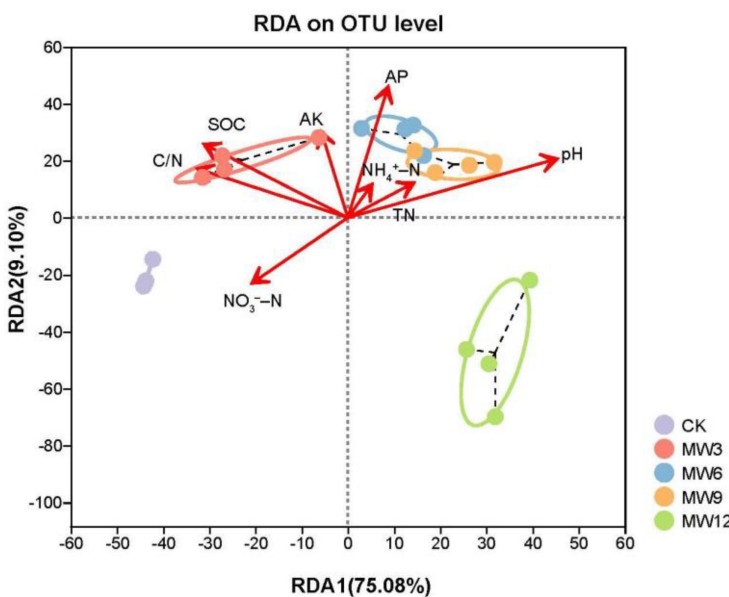

**Figure 7.** Redundancy analysis (RDA) of soil bacterial communities of different microwave treatments with environmental factors.

Soil bacterial diversity and community composition were correlated with soil physicochemical properties, as shown in Figure 8. Soil pH and AP were positively correlated with the Shannon, Chao, and Ace indices. Soil pH was significantly positively correlated with the relative abundance of Firmicutes, Actinobacteria, and *Fictibacillus* and negatively correlated with the relative abundance of Proteobacteria, Chloroflexi, Acidobacteria, *Bacillus*, *Micromonospora*, and *Paenibacillus*. SOC was significantly positively correlated with *Streptomyces*, *Micromonospora*, and *Paenibacillus*. Soil C/N was significantly and negatively correlated with the relative abundance of Firmicutes and *Fictibacillus* and significantly positively correlated with the relative abundance of Proteobacteria, Chloroflexi, *Bacillus*, *Streptomyces*, *Micromonospora*, and *Paenibacillus*. $NO_3^-N$ was significantly negatively correlated with the abundance of Actinobacteria and positively correlated with Acidobacteria. There was no significant correlation between TN, $NH_4^+$-N, and AK and soil bacterial diversity and community composition. In conclusion, soil pH, C/N, and AP may be important factors affecting bacterial community diversity, and soil bacterial community composition post microwave irradiation was driven by soil chemical properties such as soil pH, SOC, C/N, and $NO_3^-N$.

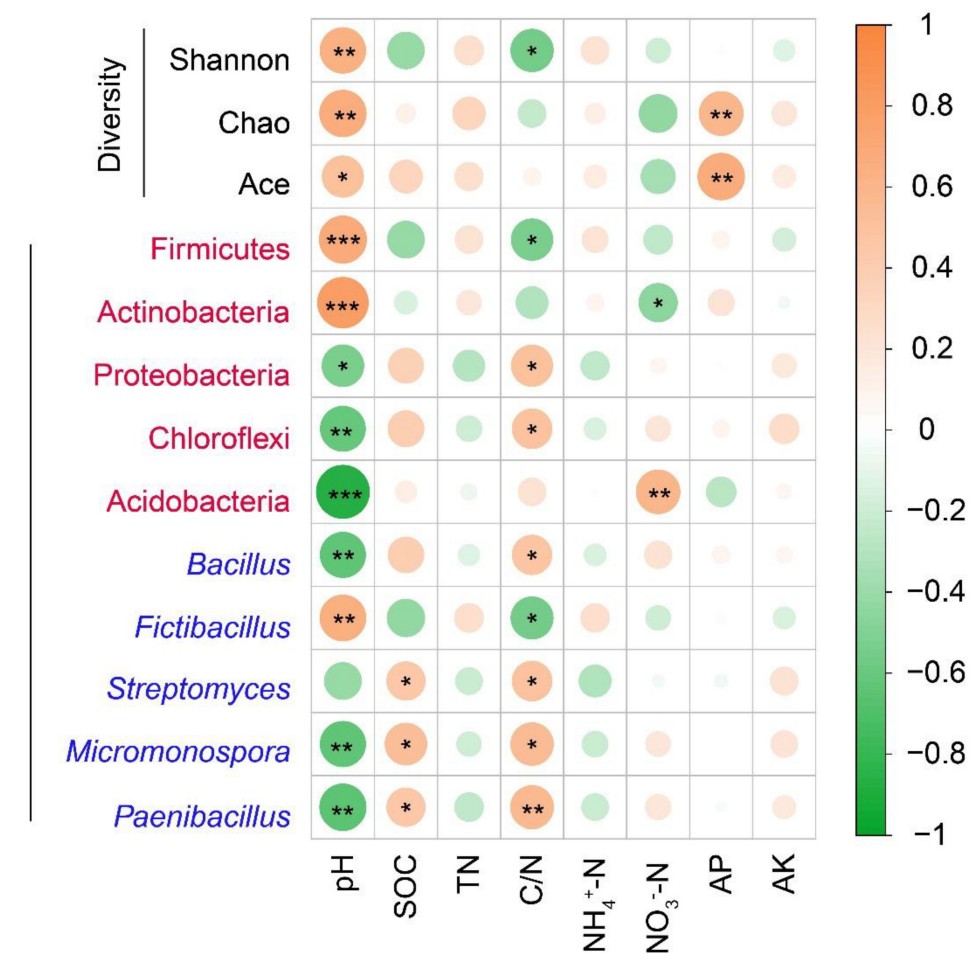

**Figure 8.** Correlation analysis between environmental factors and soil bacterial diversity at the phylum and genus level. *** $p < 0.001$, ** $p < 0.01$, * $p < 0.05$. The red regular font represents the bacterial phyla. The blue italic font represents the bacterial genera.

## 4. Discussion

### 4.1. Effect of Different Microwave Time Periods on Soil Physicochemical Properties

Soil microbial diversity and community composition are directly related to environmental factors [53,54]. Based on RDA and Mantel analysis, soil pH, C/N, and AP may be important factors influencing bacterial community diversity. Soil bacterial community composition was driven by soil chemical properties such as pH, SOC, C/N, and $NO_3^-N$ (Figure 8). The pH is considered to be a key factor affecting microbial diversity and community structure, and it has been found that a decrease in the relative abundance of bacteria and fungi may be related to pH [55–57]. This is consistent with the results of our study, where RDA analysis revealed that pH was one of the primary factors affecting microbial community structure. Soil PH plays an important role in membrane-bound proton pumping and protein stability directly imposing physiological constraints on microorganisms; when soil pH drops outside a certain range (ecological niche), it reduces the net growth of non-viable individual taxa and may alter competitive outcomes and reduce microbial diversity [58,59]. pH changes can affect bacterial community composition due to different microbial preferences, with some microorganisms being acidophilic and others alkalophilic.

The microwave-irradiated soils showed an increased mineral N ($NO_3^-$-N and $NH_4^+$-N) and AP content. Compared to the CK treatment, the MW3, MW6, MW9, and MW12 treatments increased $NH_4^+$-N content by 115.3%, 58.0%, 12.6%, and 66.8%, and AP by 19.7%, 13.4%, 12.0%, and 5.3%, respectively. MW3 and MW12 treatments increased $NO_3^-$-N content by 13.7% and 7.6%, respectively ($p < 0.05$, Table 3) [60]. The microwave irradiation-induced mechanical effects promote the diffusion of inorganic colloids, and this stimulation

increases the decomposition of non-biomass organic matter in the soil. The increase in $NO_3^-$-N may be due to the transient heat-mediated mineralization of organic nitrogen to ammonia ($NH_3$) during soil microwaving, which is converted to nitrate ($NO_3^-$) in the soil via nitrification [61]. The increase in soil $NH_4^+$-N after microwave treatment may be attributed to the fact that microwave irradiation-induced thermal denaturation of biomolecules can increase the concentration of free amino acids and then convert them to $CO_2$ and $NH_4^+$ [62]. The increase in AP content may be due to the outer layer of closed-state phosphorus (O-P) in the soil that is wrapped by an iron oxide colloid film, wherein the release of effective phosphorus is difficult [63]. During the microwave irradiation process, electromagnetic fields and thermal effects promote the diffusion of inorganic colloids, thus leading to AP release [61]. In addition, the effectiveness of phosphorus varies with soil pH; when the pH is between 6 and 7, phosphorus effectiveness is at its maximum. This study's soil pH range was between 6.9 and 7.1; therefore, it was more favorable for effective phosphorus in the soil [64]. $NH_4^+$-N, $NO_3^-$N, and AP were maintained at high levels for 30 days after microwave irradiation, indicating that microwave irradiation promoted the release of soil fast-acting nutrients, which are essential for soil fertility enhancement and crop growth [65].

*4.2. Effect of Different Microwave Time Periods on Soil Bacterial Communities*

Soil microbes play a key role in the ecosystem, driving important ecosystem processes to maintain crop productivity and species richness [66]. In our study, after 30 days of recovery from microwave irradiation, the microwave periods with the least reduction of bacterial diversity were 3 min and 6 min, which had the same bacterial diversity as the unirradiated soil ($p < 0.05$, Figure 4). Bacterial diversity decreased by 38–52% after 12-min microwave irradiation ($p < 0.05$, Figure 4), indicating that prolonged irradiation had a negative impact on soil microbial diversity and may affect soil ecosystem functions, such as nutrient cycling, carbon storage, and pest and disease control. Loss of these essential soil functions can lead to limitations in crop growth and yield [67–69]. The longer the microwave irradiation time, the higher the soil temperature and the more disturbed the microbial community. The reduction in bacterial diversity due to high temperatures is unavoidable during the microwave process [70]. Thus, precise inactivation of specific pathogenic bacteria for different plots may require a more accurate microwave irradiation time.

Microbial communities were not only affected by soil physicochemical properties, but also by their physiological characteristics that differed in their tolerance response to microwaves. For example, Actinobacteria are highly sensitive to heating, while Proteobacteria show moderate tolerance and Firmicutes show the highest tolerance [67]. For the analysis of soil bacterial community composition, it was found that the relative abundance of Actinobacteria decreased by 2.1%, 35.5%, 43.1%, and 82.0% after microwave irradiation at 3, 6, 9, and 12 min, respectively, compared with that in the CK; the abundance of Proteobacteria decreased by 31.3%, 67.6%, 78.6%, and 55.7%; the abundance of Firmicutes increased by 28%, 58%, 69%, and 83%, respectively (Figure 6a). According to the perturbation hypothesis, species exhibit a trade-off strategy when resisting certain disturbances and competition for resources; in this study, the relative abundance of Firmicutes increased while the relative abundance of Actinobacteria and Proteobacteria decreased, suggesting a trade-off between bacterial species as a result of microwave irradiation [71]. Firmicutes are copiotrophic taxa with a high degradation capacity and metabolic activity for degrading insoluble compounds in the soil and controlling plant diseases and insects [72–74]. The relative abundance of *Fictibacillus* increased with microwave irradiation duration (Figure 6b), suggesting that *Fictibacillus* was also a thermotolerant strain. *Fictibacillus* has phosphorus removal and insect resistant properties, and can produce indole-3-acetic acid, which is involved in one of the most important mechanisms for plant growth promotion by plant inter-rhizobacteria and root endophytes [75]. Formulations containing *Fictibacillus* can effectively control fungal soil-borne diseases of plants by root irrigation and reduce the occurrence of bacterial and fungal diseases on plant foliage [76]. Moreover, *Fictibacillus* has shown strong tolerance

to drought stress and biomass during the growth cycle, which can provide a high-quality microbial resource for bioremediation of soils in arid regions resulting in improvements of plant drought tolerance. The relative abundance of *Micromonospora* in soil was increased by 1.2–5-fold after microwave irradiation (Figure 6b). Previous reports have revealed that *Micromonospora* regulates crop protection by producing antifungal and antimicrobial compounds [77]. The relative abundance of *Paenibacillus* in soil after microwave irradiation was 1.6–4.9 times higher than that of CK treatment (Figure 6b). *Paenibacillus* is not only an ideal agent for controlling soil-borne diseases such as cyanosis, wilt, root rot, and soft rot, but also can promote crop growth, increase crop yield, and improve crop quality [78–80]. *Paenibacillus* is an important inter-rhizosphere promoting bacteria that can directly promote plant growth through mechanisms such as nitrogen fixation, hormone production, iron carrier secretion, and activation of mineral nutrients; in soil-borne disease control, it can resist plant diseases through mechanisms such as induction of plant disease resistance and production of various antimicrobial active substances [81–83]. The field control rate of bacterial wilt on pepper, tobacco, potato, and ginger was more than 70%, the control effect of bacterial wilt on Solanaceae crops was more significant, and the partial yield increase rate was up to 493% [84,85]. Moreover, it was shown that treatments with a high relative abundance of *Paenibacillus* increased plant height in the field by 10–30 cm compared to the control group, promoting plant development while increasing yield by ~ 27.5% [86–89]. The elevated relative abundance of *Paenibacillus* by microwave irradiation is equivalent to a certain degree of strengthening this "fungus for fungus" approach to soil-borne disease control, providing an effective means of disease control for safe, green, and organic crop production.

## 5. Conclusions

In this study, soil OTU numbers were reduced by 5.3%, 17.9%, 27.2%, and 50.1% after 30 days of microwave treatments (MW3, MW6, MW9, and MW12, respectively), compared to that in the CK. Soil bacterial diversity (Shannon, Chao, and Ace indices) decreased significantly with increasing microwave time periods, and the community structure of soil bacteria varied significantly among treatments at different microwave time periods. The relative abundance of Firmicutes increased by 28%, 58%, 69%, and 83% compared to that in the CK after 3, 6, 9, and 12 min of microwave irradiation, respectively; the relative abundance of *Bacillus, Fictibacillus, Micromonospora,* and *Paenibacillus* all showed different levels of increase compared to that in the CK treatment under the four microwave time periods. The correlation of dominant bacterial phyla and genera with soil chemical properties and diversity indices revealed that soil pH was significantly positively correlated with the relative abundance of Firmicutes, Actinobacteria, and *Fictibacillus* and negatively correlated with the relative abundance of Proteobacteria, Chloroflexi, Acidobacteria, *Bacillus, Micromonospora,* and *Paenibacillus*. Bacterial diversity was reduced by 38–52% in 9 and 12 min microwave irradiation, which implies that longer irradiation periods have a negative impact on soil microbial diversity. Microwave irradiation periods of 3 and 6 min show a smaller reduction in bacterial diversity and may be more suitable for microwave soil disinfection. The soil pH, C/N, and AP might be essential factors contributing to bacterial community diversity, and bacterial community composition after microwave soil irradiation was influenced by soil chemical properties such as soil pH, SOC, C/N and $NO_3^-$-N. In summary, microwave irradiation can help to increase the relative abundance of some beneficial bacteria in the soil, serving as a green soil disinfection method that addresses soil-borne diseases. This approach is non-polluting and can be used as part of safe, green, and organic production practices, with positive effects including improving crop quality and yield and reducing soil-borne diseases.

**Author Contributions:** Conceptualization, C.Z. (Chunjiang Zhao) and X.S.; Methodology, X.S., C.Z. (Changyuan Zhai), S.Y., C.Z.(Chunjiang Zhao), C.G.; Validation, S.Y.; Formal analyses, S.Y., Q.L.; Investigation, X.S., C.G.; Resources, C.Z. (Chunjiang Zhao), C.Z. (Changyuan Zhai); Data curation, X.S., Q.L.; Writing—original draft, X.S., Q.L.; Writing—review & editing, C.Z. (Chunjiang Zhao),

C.Z. (Changyuan Zhai), S.Y.; Funding acquisition, C.Z. (Chunjiang Zhao), C.Z. (Changyuan Zhai); Supervision, C.Z. (Chunjiang Zhao), C.Z. (Changyuan Zhai). All authors have read and agreed to the published version of the manuscript.

**Funding:** The support was provided by (1) Jiangsu Province key Research and Development Program project (BE2021302); (2) Strategic Pilot Science and Technology Program of Chinese Academy of Sciences (XDA28090108); (3) Jiangsu Province Agricultural Science and technology independent Innovation fund project (CX (21) 2006); and (4) Reform and Development Project topic (None).

**Data Availability Statement:** Not applicable.

**Conflicts of Interest:** The authors declare no conflict of interest.

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
