# Peer review of "Effect of Microwave Treatment at 2.45 GHz on Soil Physicochemical Properties and Bacterial Community Characteristics in Phaeozems of Northeast China"

_agronomy, doi:10.3390/agronomy13020600_

Round 1

Reviewer 1 Report

The overall quality of this revised paper may meet the journal's requirements.

1. In the paper, the same word is abbreviated in some places but not in others. please check carefully. e.g. line 392-394

2. add the necessary citations. e.g. Line 390-392 add a citation 

3. I read the manuscript carefully. I really feel that the author should improve the quality of the discussion. They can find more exciting points behind the research results.

Author Response

Article Title: “Effect of microwave treatment at 2.45 GHz on soil physicochemical properties and bacterial community characteristics in Phaeozems of Northeast China”

Dear Reviewers,

On behalf of all the authors, we sincerely appreciate your valuable comments on the manuscript. Your comments not only provided constructive suggestions for improving the quality of the manuscript but also led us to consider our approaches in detail. Our future research will benefit from these comments, as well.

Best regards,

Xiaohe Sun, Qi Li ,Chunjiang Zhao, Shuo Yang, Chenchen Gu, Changyuan Zhai*

Comment # 1: In the paper, the same word is abbreviated in some places but not in others. please check carefully. e.g. line 392-394.

Author response 1: Thank you for the suggestion. Based on your comments, I have corrected all non-abbreviated parts uniformly.

Comment # 2: Add the necessary citations. e.g. Line 390-392 add a citation .

Author response 2: Thank you for the suggestion. Based on your comments, I have added the relevant references in the manuscript.

Comment # 3:  I read the manuscript carefully. I really feel that the author should improve the quality of the discussion. They can find more exciting points behind the research results.

Author response 3: Thank you for the suggestion. Based on your comments,I have added relevant content such as: description of how pH affects microbial diversity and community structure based on RDA analysis in the discussion section, in lines 414-420 of the manuscript."Soil PH plays an important role in membrane-bound proton pumping and protein stability directly imposing physiological constraints on microorganisms; when soil pH drops outside a certain range (ecological niche), it reduces the net growth of non-viable individual taxa and may alter competitive outcomes and reduce microbial diversity [58,59]. pH changes can affect bacterial community composition due to different microbial preferences, with some microorganisms being acidophilic and others alkalophilic."

Reviewer 2 Report

Development of methods for soil disinfection has high relevance for the practice. Microwave irradiation can be a viable option for this purpose. But, detailed analyses are needed to investigate and analyse the effects of high power electromagnetic field at microwave frequencies on microbial community and physico-chemical properties of soil. Therefore, the topic of the manuscript can be considered as interesting and relevant. For the experiments a specific tailor made microwave apparatus was used. This experimental set-up and specific conditions provide more useful data and information that of obtained from ’household’ type microwave equipment. The manuscript is properly structured. Introduction section summarizes well the theoretical background and relevance of the study.  Applied analytical, sequencing and statistical methods are adequate and described clearly. The manuscript contains interesting and valuable results that are represented well in tables and figures and discussed in details with relevant references.

Comments, suggestions:

Please provide more detailed data and information related to the microwave equipment (geometry, power etc.).

In the Introduction section please discuss briefly the role of dielectric parameters in the efficiency and effects of microwave irradiation.

Please compare the effects and efficiency of microwave soil irradiation to conventional thermal/or conventionally used) methods.

Please discuss the practical applicability and the economy of the microwave treatment (its cost vs. other methods) in more details.

In the research irradiation frequency of 2.45 GHz was used. Lower frequency can result higher penetration depth (and more homogenous temperature, for instance). Please explain how was the frequency of microwave irradiation selected.

Author Response

Article Title: “Effect of microwave treatment at 2.45 GHz on soil physicochemical properties and bacterial community characteristics in Phaeozems of Northeast China”

Dear Reviewers,

On behalf of all the authors, we sincerely appreciate your valuable comments on the manuscript. Your comments not only provided constructive suggestions for improving the quality of the manuscript but also led us to consider our approaches in detail. Our future research will benefit from these comments, as well.

Best regards,

Xiaohe Sun, Qi Li ,Chunjiang Zhao, Shuo Yang, Chenchen Gu, Changyuan Zhai*

Comment # 1: Please provide more detailed data and information related to the microwave equipment (geometry, power etc.).

Author response 1: Thank you for the suggestion. Based on your comments, I have added the microwave soil irradiation test bench main equipment and materials and parameters related information, including the equipment (material) specifications (model) number and manufacturer, etc. in the Materials and methods section 175-176 line.

Comment # 2: In the Introduction section please discuss briefly the role of dielectric parameters in the efficiency and effects of microwave irradiation.

Author response 2: Thank you for the suggestion. Based on your comments, I have added a discussion of dielectric parameters in microwave soil irradiation in lines 63-69 of the Introduction section,"The microwave soil disinfection process can lead to dielectric heating of water in the soil and organism’s cells, and the value of the dielectric properties of soil varies depending on frequency, water content, density, and temperature [6–8]. When soil water content is the only variable, microwave action at deeper locations within the soil may become more difficult as soil water content and the loss factor increases, so a suitable soil water content tends to be more favorable for the microwave process. "

Comment # 3: Please compare the effects and efficiency of microwave soil irradiation to conventional thermal/or conventionally used) methods.

Author response 3: Thank you for the suggestion. Based on your comments, I have added a comparison between microwave soil disinfection and conventional soil disinfection in lines 55-63 of the Introduction section."Physical methods of soil disinfection such as the flame method, hot water method, and steam method mainly work through the thermal effect of denaturing proteins and inactivating pathogenic bacteria in the soil [4]. Unlike conventional heating for soil disinfection, microwave soil disinfection involves rapid oscillation of the polar molecules due to electromagnetic fields which impacts polar macromolecular traits such as changing protease activity, affecting the charge distribution near the cell membrane of the organism, causing damage to the cell membrane producing membrane dysfunction, and interfering with or destroying normal cellular metabolic functions, leading to growth inhibition, cessation, or death of the organism [5]."

Comment # 4: Please discuss the practical applicability and the economy of the microwave treatment (its cost vs. other methods) in more details.

Author response 4: Thank you for the suggestion. Microwave soil disinfection has not yet been put into production and use on a large scale, but it is worth mentioning that microwaves as an alternative to chemical agents have the advantage of being non-toxic and residue-free, as discussed in lines 49-54 line of the Introduction section of the manuscript.

Compared with traditional physical soil disinfection methods, microwave soil disinfection has a special "biological effect" both non-thermal effect."Unlike conventional heating soil disinfection, microwave soil disinfection process due to the role of electromagnetic field polar molecules oscillate rapidly, changing the polar macromolecular properties such as changing the activity of proteases, affecting the charge distribution near the cell membrane of the organism, so that the cell membrane is damaged, resulting in membrane dysfunction, interference or destruction of the normal metabolic function of the cell, resulting in the inhibition of growth, stopping or death of the organism." I have added the relevant description in lines 57-63of the Introduction section of the manuscript.

In summary, soil disinfection by microwave has good prospects, but the actual applicability and economics need to be judged in combination with production, promotion, statistics and investment in more relevant trials.

Comment # 5: In the research irradiation frequency of 2.45 GHz was used. Lower frequency can result higher penetration depth (and more homogenous temperature, for instance). Please explain how was the frequency of microwave irradiation selected.

Author response 5: Thank you for the suggestion. There is no doubt that a lower frequency can lead to a larger wavelength and may achieve a deeper penetration depth, but the size of the microwave energy is an important factor in the increase in soil temperature, 2450MHZ, 915MHZ Pupu is suitable for industrial agriculture and households in China, from the consideration of soil warming capacity, if the soil is warmed to the same temperature, 2450MHZ may be a more suitable choice for this issue. I have also added the explanation of the choice of 2.45 GHz frequency to the 148-152 line Materials and Methods.

Reviewer 3 Report

Review of Effect of microwave treatment at 2.45 GHz on soil physicochemical properties and bacterial community characteristics in Phaeozems of Northeast China

I have completed my review of manuscript Agronomy-2198565, entitled, Effect of microwave treatment at 2.45 GHz on soil physicochemical properties and bacterial community characteristics in Phaeozems of Northeast China.”

Microwave irradiation is a new means of non-toxic, residue-free, and green soil disinfection that prevents and controls soil diseases, insects, and weeds and helps to improve crop quality and yield. Soil microorganisms, as an important part of the ecosystem, are closely related to crop growth and health. To investigate the changes in soil physicochemical properties and microbial communities during microwave soil disinfection for different periods. Phaeozems from northeastern China were selected for microwave treatment at 3, 6, 9, and 12 min, and their physicochemical properties were measured after 30 days of incubation. According to the authors, microwave treatments at different periods affected soil microbial community structure to different degrees, and soil bacteria of copiotrophic taxa (e.g., Firmicutes) were relatively higher than the control. Overall, microwave treatment from 3–6 min may be more suitable for soil disinfection. The study of the effect of microwaves on soil physicochemical properties and bacterial microbial community not only provides some scientific reference for the rational application of microwave soil disinfection but also has positive significance for soil-borne disease control and crop quality improvement.

The subject and findings of this article are interesting and useful. Before making a positive decision, I have some concerns and comments about the present form of the manuscript that must be addressed first.

Comments for authors

Comment 1: I advise the authors to include a section in the introduction explaining the mechanism by which microwave radiation affects the biological system to provide more background information. The suggested article might be helpful for authors to explain and strengthen the background details.

Article: Microwave Radiation and the Brain: Mechanisms, Current Status, and Future Prospects. International Journal of Molecular Sciences vol. 23 (2022). [https://doi.org/10.3390/ijms23169288].

Comment 2: In the introduction, increased the introduction related to Phaeozems.

Comment 3: The results are depended on the properties of the microwave, which is a crucial factor for research and real application. I believe this article is missing information about microwaves and their characteristics. The material and method section should include a thorough explanation of the microwave exposure system.

Comment 4: What is the microwave type? Pulsed or continues? How much is the power? Add these details to the article.

Comment 5: How authors measured the temperature of the soil and penetration depth after exposure?

Comment 6: Explain the findings in line number 213 of Table 2 where the pH, nitrate, and ammonium show an odd trend after increasing microwave treatment. To make the OTU clear to the readers, the results section also included an explanation in line number 231.

Comment 7: There is little to no difference between microwave treatment and conventional treatment. You must describe how pH is the primary factor affecting microbial diversity and community structure in terms of RDA analysis in the discussion (lines 381 to 384).

Comment 8: The paper contains errors and typos that make it difficult to understand and distort its intended meaning. I encourage authors to reread carefully and fix any grammatical errors.

In my opinion, the manuscript should be placed under major revisions.

Author Response

Article Title: “Effect of microwave treatment at 2.45 GHz on soil physicochemical properties and bacterial community characteristics in Phaeozems of Northeast China”

Dear Reviewers,

On behalf of all the authors, we sincerely appreciate your valuable comments on the manuscript. Your comments not only provided constructive suggestions for improving the quality of the manuscript but also led us to consider our approaches in detail. Our future research will benefit from these comments, as well.

Best regards,

Xiaohe Sun, Qi Li ,Chunjiang Zhao, Shuo Yang, Chenchen Gu, Changyuan Zhai*

Comment # 1: I advise the authors to include a section in the introduction explaining the mechanism by which microwave radiation affects the biological system to provide more background information. The suggested article might be helpful for authors to explain and strengthen the background details.

Author response 1: Thank you for the suggestion, Based on your comments, I have added the explanation of the "Mechanistic effects of microwave irradiation on biological systems" in lines 55-69 of the Introduction,"Physical methods of soil disinfection such as the flame method, hot water method, and steam method mainly work through the thermal effect of denaturing proteins and inactivating pathogenic bacteria in the soil [4]. Unlike conventional heating for soil disinfection, microwave soil disinfection involves rapid oscillation of the polar molecules due to electromagnetic fields which impacts polar macromolecular traits such as changing protease activity, affecting the charge distribution near the cell membrane of the organism, causing damage to the cell membrane producing membrane dysfunction, and interfering with or destroying normal cellular metabolic functions, leading to growth inhibition, cessation, or death of the organism [5].The microwave soil disinfection process can lead to dielectric heating of water in the soil and organism’s cells, and the value of the dielectric properties of soil varies depending on frequency, water content, density, and temperature [6–8]. When soil water content is the only variable, microwave action at deeper locations within the soil may become more difficult as soil water content and the loss factor increases, so a suitable soil water content tends to be more favorable for the microwave process. "

Comment # 2: In the introduction, increased the introduction related to Phaeozems.

Author response 2: Thank you for the suggestion. Based on your comments,I have added a description of the properties and characteristics of phaeozems(black soil) in lines 120-125 of the Introduction,"Phaeozems are clayey soils with strong expansion and contraction and disturbance characteristics. Phaeozems in northeast China are located in one of the world's major phaeozems belts [41], which are equivalent to the US classification of metamorphic soils and the UN classification of metamorphic soil units; this kind of soil has good natural conditions and a high fertility, making it very suitable for agricultural farming [42]."

Comment # 3: The results are depended on the properties of the microwave, which is a crucial factor for research and real application. I believe this article is missing information about microwaves and their characteristics. The material and method section should include a thorough explanation of the microwave exposure system.

Author response 3: Thank you for the suggestion. Based on your comments, I have added the microwave soil irradiation test bench main equipment and materials and parameters related information, including the equipment (material) specifications (model) number and manufacturer, etc. in the Materials and methods section 175-176 line Table 1.

Comment # 4: What is the microwave type? Pulsed or continues? How much is the power? Add these details to the article.

Author response 4: Thank you for the suggestion. The type of microwave is 2.45 MHz continuous output with a power of 2000 W. I have added the specific description to the Materials and Methods section in lines 148-152.

Comment # 5: How authors measured the temperature of the soil and penetration depth after exposure?

Author response 5: Thank you for the suggestion. The internal temperature of the soil after microwave treatment was measured by a homemade PT100 sensor (Figure 1), which was inserted vertically into the soil to measure each cm of soil temperature in turn. And I have added in lines 152-155.

Comment # 6: Explain the findings in line number 213 of Table 2 where the pH, nitrate, and ammonium show an odd trend after increasing microwave treatment. To make the OTU clear to the readers, the results section also included an explanation in line number 231.

Author response 6: Thank you for the suggestion. Based on your comments, The reasons for the changes in pH, NO3--N, etc. have been explained in subsection 4.1 of the Discussion section. We also have added an explanation of OUT in the manuscript in lines 260-262,"OTU is a uniform marker artificially assigned to a taxonomic unit (strain, genus, species, group, etc.) in population genetics studies."

Comment # 7: There is little to no difference between microwave treatment and conventional treatment. You must describe how pH is the primary factor affecting microbial diversity and community structure in terms of RDA analysis in the discussion (lines 381 to 384).

Author response 7: Thank you for the suggestion. Based on your comments, I have describe how pH is the primary factor affecting microbial diversity and community structure in terms of RDA analysis in the discussion in line 414-420,"Soil PH plays an important role in membrane-bound proton pumping and protein stability directly imposing physiological constraints on microorganisms; when soil pH drops outside a certain range (ecological niche), it reduces the net growth of non-viable individual taxa and may alter competitive outcomes and reduce microbial diversity [58,59]. pH changes can affect bacterial community composition due to different microbial preferences, with some microorganisms being acidophilic and others alkalophilic."

Comment # 8: The paper contains errors and typos that make it difficult to understand and distort its intended meaning. I encourage authors to reread carefully and fix any grammatical errors.

Author response 8: Thank you for the suggestion. Based on your comments, I have once again touched up the article and fixed some of the grammatical errors.

Round 2

Reviewer 3 Report

The authors address my comments and concerns in the revised version. I recommend accepting the paper in its present form.